# Impact of Fertilizer on Crop Yield and C:N:P Stoichiometry in Arid and Semi-Arid Soil

**DOI:** 10.3390/ijerph18084341

**Published:** 2021-04-20

**Authors:** Qiang Liu, Hongwei Xu, Haijie Yi

**Affiliations:** 1College of Resources and Environmental Engineering, Tianshui Normal University, Tianshui 741000, China; liuqiang192@mails.ucas.ac.cn; 2State Key Laboratory of Soil Erosion and Dryland Farming on Loess Plateau, Institute of Soil and Water Conservation, Chinese Academy of Sciences and Ministry of Water Resources, Yangling 712100, China; haijieyi@nwafu.edu.cn; 3University of Chinese Academy of Sciences, Beijing 100049, China

**Keywords:** C:N:P stoichiometry, crop yield, organic fertilizer, inorganic fertilizer, long-term fertilization, arid and semi-arid soil

## Abstract

Fertilization can significantly affect the quality of crop and soil. To determine the effects of long-term fertilization on crop yield and carbon:nitrogen:phosphorus (C:N:P) stoichiometry in soil, a study was conducted on the terraced fields of the Loess Plateau from 2007 to 2019. Nine fertilization treatments were included: no fertilizer; organic fertilizer (O); organic and nitrogen fertilizers (ON); organic, nitrogen, and phosphorus fertilizers (ONP); organic and phosphorus fertilizers (OP); phosphorus and nitrogen fertilizers; potash and nitrogen fertilizers; potash, nitrogen, and phosphorus fertilizers; and potash and phosphorus fertilizers. Under these treatments except for CK and PK, crop yields initially decreased but later increased. The nutrient content and C:N:P stoichiometry increased in soil depth of 0–20 cm. The soil available nutrients did not change significantly with the duration of fertilization. The O, ON, ONP, and OP had the most evident effect on the enhancement of soil nutrient content, whereas O and ON had the most evident effect on the increase in soil organic carbon (SOC):total phosphorus (TP) and total nitrogen (TN):TP. In soil depth of 0–20 cm, crop yield, SOC:TN, SOC:TN, SOC:TP, and TN:TP significantly correlated with soil nutrients. This study indicated that long-term fertilization can effectively improve crop yield, soil fertility, and soil C:N:P stoichiometry. Meanwhile, the single application of an organic fertilizer or the combination of organic and nitrogen fertilizers can improve the condition of nitrogen limitation in arid and semi-arid areas.

## 1. Introduction

Ecological stoichiometry is the science of studying the energy balance of biological systems and multiple nutrient elements (mainly, carbon (C), nitrogen (N), and phosphorus (P)) [1]. It provides a new comprehensive method for studying plant–soil interactions and C, N, and P cycles [1,2]. Cropland ecosystems are one of the most frequently disturbed ecosystems by human activities. Soil is an important part of the ecosystem and the basis for plant growth. C, N, P, and other nutrient elements are important factors for plant growth and development. They directly affect plant growth and development, soil microbial activities, and soil nutrient cycling [3].

In agricultural production practices, fertilizers (organic and inorganic) are used to increase the accumulation of C, N, and P in the soil to improve the soil’s ability to supply nutrients to crops [4]. Fertilizers have considerably contributed to increase in food production and food security [5,6,7]. However, there are problems with high frequency of fertilizer application and low utilization efficiency [8,9]. The use of large amounts of chemical fertilizers has caused a series of harmful ecological and environmental impacts, such as soil acidification, soil compaction, and soil fertility degradation, which severely restrict the sustainable development of green agriculture [9,10]. Therefore, rationally balancing chemical fertilizer input and enhancing crop yields and soil sustainability are the targets that need to be focused urgently in current agricultural production process.

Fertilization is one of the most important methods in agricultural production. Studies have found that soil nutrients and crop growth are affected by frequent fertilization and other factors [9,11]. Fertilization leads to changes in crop yield and soil nutrient content [6,12]. Luo et al. (2020) [13] reported that fertilization significantly increased the organic and activated carbon contents in soil but reduced the soil organic carbon (SOC):total nitrogen (TN) and SOC:total phosphorus (TP) ratios; however, there were considerable differences among different fertilization methods. A study has reported that excessive fertilization reduces the overall soil quality [13]. Additionally, a study has reported that fertilization affects soil nutrient content and crop yield by affecting the ability of crops to absorb and distribute elements, such as N and P [12]. Studying the changes in characteristics of soil C, N, and P and their stoichiometric ratios under long-term fertilization conditions can help us understand in detail the status of soil nutrient cycling during plant growth, and the use of resources by plants during fertilization [14,15]. At the same time, studying the changes in soil stoichiometry under long-term fertilization can help us understand nutrient cycling and nutrient limitation in the soil ecosystem [16]. It is of great significance to understand the sustainability of soil quality and crop yield after long-term fertilization.

In the present study, we chose a typical terraced field plot in the Loess Plateau to conduct a 12-year experiment (from 2007 to 2019) on various fertilization treatments. The objective was to study the changes in characteristics of crop yield, soil nutrient content, and C:N:P stoichiometry with various fertilization methods under different fertilization durations. We hypothesized that (1) long-term fertilization may affect the crop yield and soil nutrient content, (2) the changes in soil nutrient content may alleviate the C:N:P stoichiometry of soil, and (3) the single application of an organic fertilizer or an organic fertilizer combined with an inorganic fertilizer may have an advantage in increasing crop yield and soil nutrients supply.

## 2. Materials and Methods

### 2.1. Study Area

The study site was located at the Ansai Research Station of the Chinese Academy of Sciences (36°51 N, 109°19 E, 1068–1309 m above sea level), which is a long-term detection sampling site (Figure 1). The annual average temperature is 8.8 °C. The annual average rainfall is 500 mm, which is mainly observed in July–September. It is a typical hilly and gully area in the Loess Plateau, an ecologically fragile area in the northwest China, and a key area for returning farmland to forest (grass) [17]. The soil type is loess soil.

### 2.2. Field Experiment and Design

In the experiment, a rectangular block with an area of 1080 m^2^ was selected and divided into 36 small plots of 3.5 m × 8.57 m. In total, 9 fertilizer treatments were performed, and each treatment was repeated 4 times (Table 1). The crop rotation mode was millet → glutinous millet → millet → soybean. The crops were planted once a year without irrigation. In 2007, millet was planted first. For unification, this study was for only selected crops and soil that were planted for millet. The basic soil physical and chemical properties before sowing are shown in Appendix A.

Fertilization included nine treatments, CK, O, ON, ONP, OP, NP, NK, NPK, and PK, the details of which can be found in Table 2. Winter sheep manure was selected as the organic fertilizer, and urea, diammonium phosphate, and potassium sulfate were selected as nitrogen, phosphorus, and potash fertilizers, respectively.

Sowing was performed in May every year. Before sowing the crop seeds, all plots were irrigated to meet the soil moisture requirement for crop growth. Two days after irrigation, fertilizer was applied, the artificial soil and fertilizer were mixed evenly, and the crop seeds were sown. Irrigation was not performed during the crop growth phase, and when the crops were mature, they were harvested by artificial harvesting. After harvesting, the soil was plowed manually to prepare it for crop planting later.

### 2.3. Soil Sampling and Analysis

We collected soil samples from each plot according to the S-type sampling method [18]. The soil depths for collection were 0–20 and 20–40 cm. Before collecting the samples, the litter on the surface ground was removed. Furthermore, the soil samples from each plot were mixed evenly as one soil sample and were brought back to the laboratory [18].

First, small stones were removed from the soil samples, and the roots and small animals were visible. Furthermore, it was passed through a 2-mm sieve and air-dried naturally for the determination of soil nutrient content and pH. SOC and soil TN were determined using the H_2_SO_4_–K_2_Cr_2_O_7_ method [19] and Kjeldahl method [19], respectively. Soil TP was determined using the molybdenum blue method [18]. Soil available nitrogen (AN) was determined using the alkaline KMnO_4_ method [19], and available phosphorus (AP) was determined using the Olsen method [19].

### 2.4. Statistical Analysis

Effects of the type of fertilization treatment and duration of fertilization on crop yield, SOC, TN, TP, AN, AP, pH, SOC:TN, SOC:TP, TN:TP, and AN:AP were evaluated using two-way ANOVA. One-way ANOVA was used to evaluate the crop yield, SOC, TN, TP, AN, AP, pH, SOC:TN, SOC:TP, TN:TP, and AN:AP responses to the duration of fertilization. Duncan’s post hoc test at *P* < 0.05 was used for multiple comparisons. Before ANOVA, one sample Kolmogorov–Smirnov test and the homogeneity of variance test were performed to determine whether the parameters were normally distributed and the variances were homogeneous. If not, a logarithmic transformation was used for the corresponding parameters. A correlation matrix was used to study the correlations among crop yield, SOC, TN, TP, AN, AP, pH, SOC:TN, SOC:TP, TN:TP, and AN:AP in soil depths of 0–20 and 20–40 cm during the plantation of three succession species. The correlation matrix was visualized using R. 4.0.2 (corrplot) software.

## 3. Results

### 3.1. Crop Yield

The type of fertilization treatment and duration of fertilization had a considerable impact on crop yield (Table 3). Between 2007 and 2019, crop yield after the nine fertilization treatments initially decreased but later increased as the number of years of fertilization increased (Figure 2). For OPN, NPK, ON, NP, OP, and O treatments, the crop yield was the highest in 2019.

### 3.2. Soil Nutrients

At 0–20 cm soil depths, the duration of fertilization had a significant impact on SOC, TN, TP, AP, and pH (Table 3). Between 2007 and 2019, SOC, TN, and TP after all the fertilization treatments exhibited an overall increasing trend with the duration of fertilization (Figure 3a–c). The AN and AP showed no marked changes with the duration of fertilization, whereas the overall pH decreased (Figure 3d–f).

In soil depths of 20–40 cm, the duration of fertilization had a significant impact on SOC, TN, TP, AN, AP, and pH (Table 3). Between 2007 and 2019, the SOC initially increased and further stabilized as the duration of fertilization increased (Figure 4a–c), whereas TN, TP, AN, AP, and pH did not exhibit evident changes (Figure 4d–f).

### 3.3. Soil C:N:P Stoichiometry

In soil depths of 0–20 cm, the duration of fertilization had a significant impact on SOC:TN, SOC:TP, TN:TP, and AN:AP (Table 3). Between 2007 and 2019, SOC:TN and SOC:TP after all the fertilization treatments and TN:TP after O and ON treatments exhibited an overall increasing trend with the duration of fertilization (Figure 4a,b), whereas AN:AP did not exhibit any evident changes (Figure 5d).

In soil depths of 20–40 cm, the duration of fertilization had a significant impact on SOC:TN, SOC:TP, TN:TP, and AN:AP (Table 3). Between 2007 and 2019, SOC:TN after all the fertilization treatments exhibited an overall increasing trend with the duration of fertilization (Figure 6a); the SOC:TP initially increased and later stabilized (Figure 6b), whereas TN:TP and AN:AP did not exhibit any evident changes (Figure 6c,d).

### 3.4. The Relationship among Various Parameters

For the soil depth of 0–20 cm, the correlation matrix showed that crop yield was significantly correlated with SOC, TN, TP, AN, AP, pH, and SOC:TN; SOC:TN with SOC, TN, AN, AP, pH, SOC:TP, and TN:TP; SOC:TP with SOC, TN, TP, AN, AP, TN:TP, and AN:AP; and TN:TP with SOC, TN, TP, AN, AP, pH, and AN:AP (Figure 7a).

For the soil depth of 20–40 cm, the correlation matrix showed that crop yield was significantly correlated with SOC and TP; SOC:TN with SOC, TN, AN, AP, pH, and SOC:TP; SOC:TP with SOC, TN, TP, and TN:TP; and TN:TP with SOC, TN, TP, AN, pH, and AN:AP (Figure 7b).

## 4. Discussion

### 4.1. Long-Term Fertilization Altered Crop Yield

Increasing crop production is one of the ways to increase agricultural development in arid and semi-arid regions. Fertilization is an important agronomic measure, and appropriate application of fertilizers can promote the growth and development of farming and improve the quality and yield of crops [20,21]. This study demonstrated that after the application of organic fertilizers, inorganic fertilizers, or a combination of both, crop yield initially decreased but later increased significantly as the duration of fertilization increased.

This result can be explained as follows. First, N, P, and potassium (K) are essential nutrients for the growth and development of crops. During the long-term growth of crops, the nutrient elements in the soil are consumed in large quantities. The addition of artificial fertilizers compensates for the deficiency of soil nutrients, thereby increasing the absorption and assimilation of soil N, P, and K by plants; this affects the growth and development of crops, further affecting crop yield [22]. Second, with the extension of the duration of fertilization, the application of fertilizer gradually increases the basic soil fertility; the correlation matrix showed that the crop yield and soil nutrient content were positively correlated. When it can provide enough mineral nutrients, organic matter gradually releases nutrients through mineralization to increase crop yields. The decrease in yield in the initial stage of fertilization may be because of the effect of local climate conditions or factors such as plant growth.

Studies have reported that organic fertilizer is released slowly, and it cannot meet the growth needs of crops in the current season. However, the fertilizer effect is long [21,23]. This study demonstrated that the combination of organic and inorganic fertilizers or the combination of inorganic fertilizers have clear yield-enhancing effects. The results of this study showed that fertilization is one of the key technical measures to maintain sustainable crop productivity. A reasonable combination of fertilizers is more conducive to fertilize the soil and provide conditions for the full growth of crops. Gosal et al. (2018) [10] reported that the combined application of inorganic and organic fertilizers or mixture of inorganic fertilizers doubled the input of N, P, and K; significantly increased the nutrient content in the soil; and helped in meeting the nutrient requirement for crop growth. This resulted in a considerable increase in crop yield. The long-term addition of fertilizers can release fixed elements in soil, improve soil structure and ecological conditions, and balance the soil’s capacity of water and fertilizer supply [9]. On the contrary, long-term planting without fertilization leads to continuous nutrient deficiency in the soil, the soil becomes increasingly barren, and the nutrients will not suffice for the growth of the crop, which affects the crop yield.

### 4.2. Long-Term Fertilization Altered Soil Nutrients

Soil nutrients can provide the material basis and energy source for the survival of various microorganisms in the soil and the growth of crops [24]. Soil nutrients reflect the maturity of the soil to a certain extent [25]. This study reported that the SOC, TN, and TP content of the surface layer (0–20 cm) of soil after fertilization exhibited an overall increasing trend as the duration of fertilization increased. The increase in soil nutrients is mainly due to the input of exogenous soil nutrients. Moreover, soil microbial activity determines the intensity of the soil biochemical reaction processes and participates in the circulation and transformation of nutrients in the soil, which can reflect the status of soil fertility [26]. Studies have reported that fertilization can increase the diversity and composition of soil microorganisms, thereby producing a large number of enzymes [6,27] and improving soil fertility in turn. In addition, soil pH has a significant impact on obtaining the available nutrients from the soil by crops [28], and changes in soil pH also significantly affect the microbial activity in the soil [25,29]. This study demonstrated that long-term fertilization reduced the overall pH of the surface soil, which is consistent with the report by Chen et al. (2017) [9]. It is essential that soil has a certain pH range for the growth of crops. In this study, it was found that under fertilization, soil pH transforms to a range that is more suitable for crop growth and soil microbial growth, which is conducive to the accumulation of soil nutrients. The correlation matrix showed that soil pH is significantly correlated with SOC and TN content, which also confirms the above conclusion. In addition, although available soil nutrients can be directly used by crops as effective nutrients, in this study, we found that fertilization did not significantly change the available nutrients in soil. Fertilization had no significant effect on soil nutrient content at a soil depth of 20–40 cm. The principal reason is that the principal growth environment for the crop is the surface layer. Moreover, the crop root system is shallow; the soil structure and permeability are poor, and the environmental microbial activity is low in the lower soil layer. Therefore, fertilization has little effect on the nutrients in the deep soil. At the same time, the surface soil is greatly affected by the external environment and the return of litter nutrients; thus, the nutrients first gather in the surface soil and then migrate to the lower layer.

A lack of fertilization long term leads to an increase in the soil nutrient content in the soil layer of 0–20 cm, which is inconsistent with the conclusion that a lack of fertilization long term leads to a decrease in soil organic matter content [30]. It may be related to crop types and local environmental and climatic conditions. The soil nutrient content during the lack of fertilization long term exhibited a slightly increasing trend; the reason may be that during the entire growth cycle of the crop, its aboveground growth is vigorous, and the residual branches, leaves, and roots return to the soil through the humification process, which increases the soil nutrient content.

At a soil depth of 0–20 cm, the application of organic fertilizer or mixed application of organic and inorganic fertilizers had a more enhanced effect on soil nutrient content than the single application of inorganic fertilizer (Figure 3, Appendix A). This is because exogenous organic fertilizers can directly increase the C, N, and P content of the soil. In addition, long-term single application of inorganic fertilizers might not make the soil nutrients gradual and continuous. Although, to a certain extent, inorganic fertilizers can significantly increase crop yields. However, under the condition of no external organic fertilizer supplement, the long-term growth of crops will increase the consumption of nutrients in the soil, which will have a small increasing effect.

### 4.3. Long-Term Fertilization Altered C:N:P Stoichiometry in Soil

In this study, it was found that the SOC:TN in 0–20 cm soil layer increased after fertilization with the increase in the number of years of fertilization. Studies have found that the main factors affecting the changes in SOC:TN are changes in SOC and soil TN content after fertilization [25,31]. SOC:TN in soil has a significant positive correlation with SOC and TN (Figure 7a). In this study, the overall rate of increase in SOC was greater than that of soil TN (Appendix A). Therefore, the SOC:TN in soil increased after fertilization. SOC:TN can reflect the decomposition rate of organic matter in the soil [32]. Therefore, we believe that the decomposition rate of soil organic matter after fertilization increases with the increase in the duration of fertilization, which is beneficial to the growth of crops. In addition, the overall SOC:TN in this study was lower than the average value in China and the world [33,34]. Under normal circumstances, lower SOC:TN can promote the vitality of soil microorganisms [35] and help in enhancing the conversion of nutrients in the soil.

SOC:TP in soil can characterize the potential of soil organic matter mineralization to release or absorb and fix P [25,32]. Additionally, this study reported that after fertilization, the rate of increase in SOC was greater than that in soil TP, resulting in an increase in the SOC:TP with the increase in the number of years of fertilization (Appendix A). In addition, the SOC:TP ratios in soil under all fertilization treatments were lower than the average value in China [34]. This may be because the soil nutrients in the arid and semi-arid areas are poor, and the microbial activity is low. To maintain the growth of vegetation, the circulation of soil nutrients must be improved so that the rate of SOC:TP is higher.

In addition, we found that the overall rate of increase in TN was greater than that in TP under O and ON treatments (Appendix A), resulting in an increase in the TN:TP with increasing duration of fertilization. TN:TP is used as an index of soil nutrient restriction [34], and N and P are essential mineral nutrients and are common limiting elements for plant growth. In this study, we found that O and ON treatments had a higher TN:TP, which indicated that applying O or ON treatments can improve soil properties in a better way. However, the TN:TP in all fertilization treatments was at a low level, indicating that in arid and semi-arid regions, crop growth is limited by N for a long time. In addition, this study reported that long-term fertilization had little effect on the C:N:P stoichiometry of the 20–40 cm deep soil layer, which was mainly attributed to the lower change in SOC, TN, and TP.

## 5. Conclusions

In conclusion, we studied nine fertilization treatments for millet in arid and semi-arid areas and found that crop yield and C:N:P stoichiometric ratio in soil were significantly affected by the duration of fertilization. The long-term use of organic fertilizer, inorganic fertilizer, or a combination of both can effectively increase crop yield, improve soil nutrient environmental conditions, and improve C:N:P stoichiometry in 0–20 cm deep soil layer. The effect of organic fertilizer and combined application of organic and inorganic fertilizers on increasing soil nutrient content is more evident. Fertilization using organic fertilizer or organic fertilizer combined with nitrogen fertilizer can effectively alleviate nitrogen limitation in arid and semi-arid areas. Therefore, this study provided supportive data for changes in soil nutrient content and C:N:P stoichiometric ratio after long-term fertilization in arid and semi-arid regions and provided a theoretical basis for efficient soil fertilization in such areas.

## Figures and Tables

**Figure 1 ijerph-18-04341-f001:**
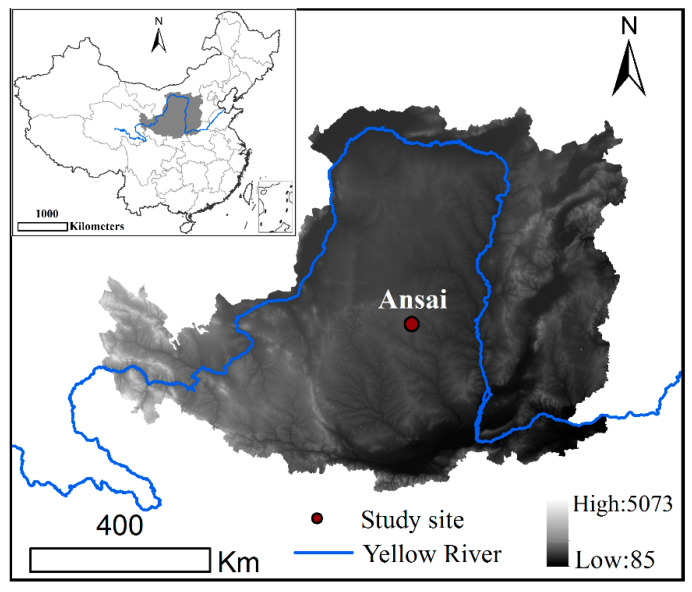
The locations of study sites in Ansai County, Shanxi Province, China.

**Figure 2 ijerph-18-04341-f002:**
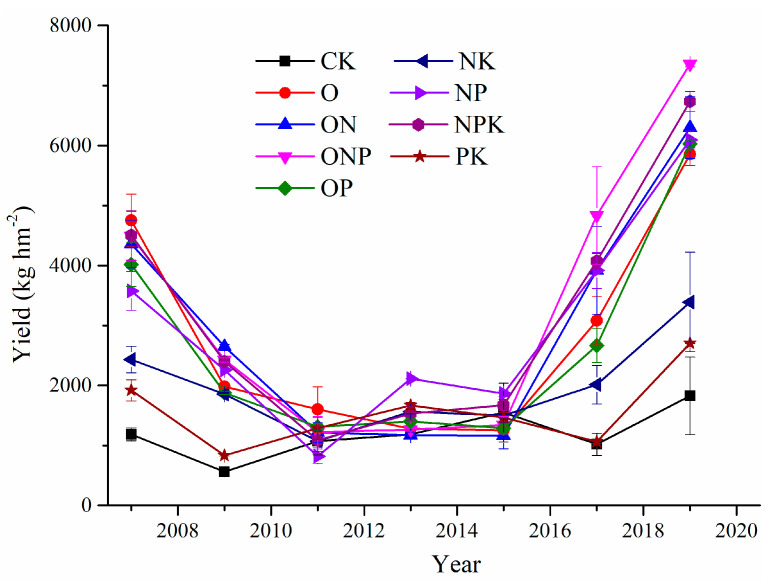
Crop yield after various fertilization treatments. Values are presented as the mean ± SE.

**Figure 3 ijerph-18-04341-f003:**
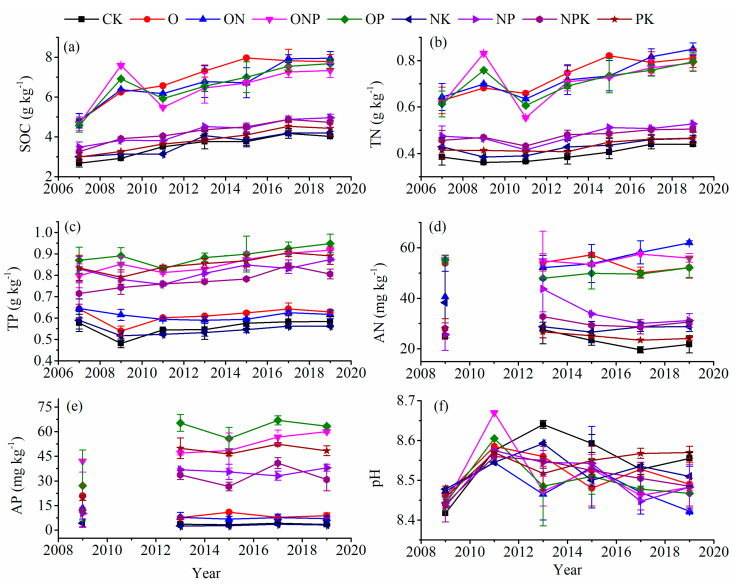
Effects on soil nutrients [SOC (**a**), TN (**b**), TP (**c**), AN (**d**), and AP (**e**)] and pH (**f**) after various fertilization treatments in soil depths of 0–20 cm. Notes: Values are presented as the mean ± SE; Soil samples used for AP and AN determination of 2009 and 2011 are missing.

**Figure 4 ijerph-18-04341-f004:**
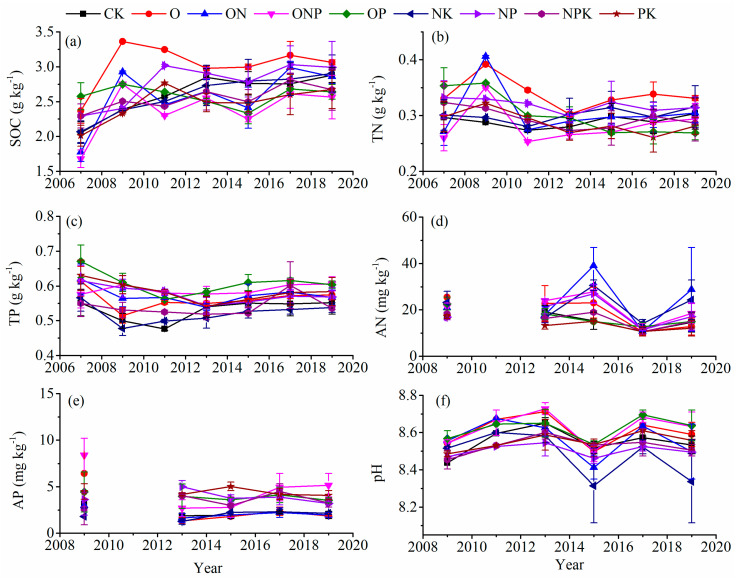
Effects on soil nutrients (SOC (**a**), TN (**b**), TP (**c**), AN (**d**), and AP (**e**)) and pH (**f**) after various fertilization treatments in soil depths of 20–40 cm. Notes: Values are presented as the mean ± SE; Soil samples used for AP and AN determination of 2009 and 2011 are missing.

**Figure 5 ijerph-18-04341-f005:**
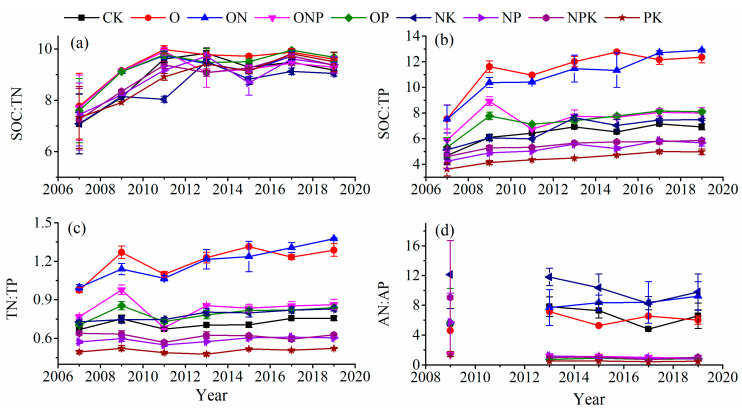
C:N:P stoichiometry (SOC:TN (**a**), SOC:TP (**b**), TN:TP (**c**), and AN:AP (**d**)) in soil after various fertilization treatments in soil depths of 0–20 cm. Notes: Values are presented as the mean ± SE; Soil samples used for AP and AN determination of 2009 and 2011 are missing.

**Figure 6 ijerph-18-04341-f006:**
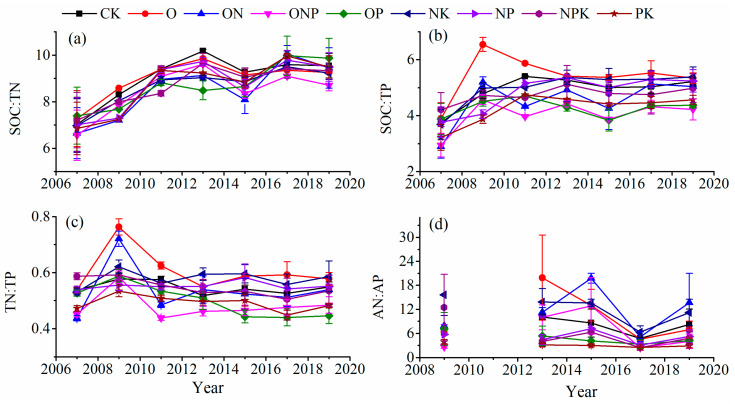
C:N:P stoichiometry (SOC:TN (**a**), SOC:TP (**b**), TN:TP (**c**), and AN:AP (**d**)) in soil after various fertilization treatments in soil depths of 20–40 cm. Notes: Values are presented as the mean ± SE; Soil samples used for AP and AN determination of 2009 and 2011 are missing.

**Figure 7 ijerph-18-04341-f007:**
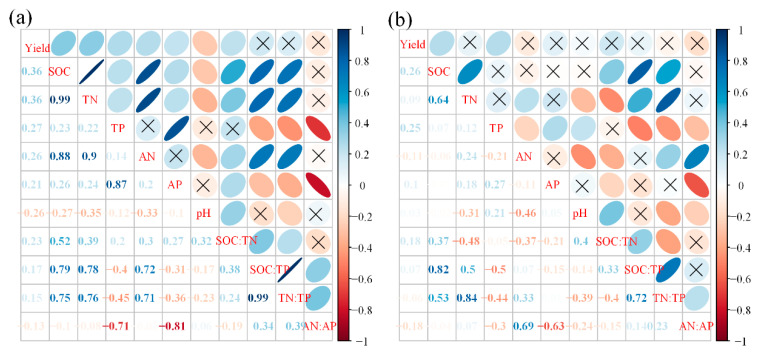
Correlation matrix among the different parameters determined for soil depths of 0–20 cm (**a**) and 20–40 cm (**b**). Notes: ×, correlation is non-significant at *P* < 0.05.

**Table 1 ijerph-18-04341-t001:** Experimental design of the different fertilization treatments.

Experimental Plots								
1 ON	2 O	3 NPK	4 PK	5 NK	6 NP	7 CK	8 PNK	9 PP
10 OP	11 ON	12 PK	13 NK	14 CK	15 NPK	16 NK	17 O	18 ONP
19 ONP	20 OP	21 CK	22 NK	23 NP	24 PK	25 NPK	26 ON	27 O
28 O	29 ONP	30 NP	31 CK	32 NPK	33 NK	34 PK	35 OP	36 ON

**Table 2 ijerph-18-04341-t002:** Experimental fertilization.

Treatment	Illustration
CK	No fertilizer
O	0.75 kg/m^2^ organic fertilizer
ON	0.75 kg/m^2^ organic fertilizer and 0.021 kg/m^2^ nitrogen
ONP	0.75 kg/m^2^ organic fertilizer, 0.021 kg/m^2^ nitrogen and 0.017 kg/m^2^ phosphorus
OP	0.75 kg/m^2^ organic fertilizer and 0.017 kg/m^2^ phosphorus
NP	0.017 kg/m^2^ phosphorus and 0.021 kg/m^2^ nitrogen
NK	0.012 kg/m^2^ potash and 0.021 kg/m^2^ nitrogen
NPK	0.012 kg/m^2^ potash, 0.021 kg/m^2^ nitrogen and 0.017 kg/m^2^ phosphorus
PK	0.012 kg/m^2^ potash and 0.017 kg/m^2^ phosphorus

**Table 3 ijerph-18-04341-t003:** F-value and P-value of year, fertilizer treatments (FT), and their interactions with various parameters studied by a two-way ANOVA.

	Indexes	Year		FT		Year × FT	
		*F*	*P*	*F*	*P*	*F*	*P*
	Yield	279.13	0.00	40.49	0.00	10.29	0.00
0–20 cm	SOC	16.73	0.00	137.24	0.00	1.14	0.34
	TN	11.07	0.00	185.63	0.00	1.24	0.25
	TP	10.53	0.00	149.99	0.00	0.51	0.98
	AN	0.39	0.81	39.33	0.00	0.85	0.68
	AP	6.93	0.00	64.29	0.00	2.30	0.00
	pH	8.41	0.00	2.80	0.01	0.96	0.54
	SOC:TN	37.90	0.00	5.97	0.00	2.04	0.01
	SOC:TP	8.22	0.00	296.55	0.00	1.63	0.07
	TN:TP	2.69	0.04	361.38	0.00	1.67	0.06
	AN:AP	1.38	0.26	13.55	0.00	0.66	0.89
20–40 cm	SOC	3.13	0.02	5.47	0.00	0.92	0.60
	TN	10.84	0.00	4.41	0.00	1.14	0.34
	TP	2.74	0.04	6.64	0.00	0.61	0.93
	AN	9.31	0.00	2.77	0.01	0.94	0.57
	AP	2.87	0.03	5.61	0.00	1.15	0.33
	pH	8.99	0.00	3.51	0.00	0.55	0.96
	SOC:TN	43.09	0.00	1.83	0.10	1.36	0.17
	SOC:TP	1.72	0.16	12.41	0.00	1.23	0.26
	TN:TP	14.57	0.00	9.69	0.00	1.10	0.38
	AN:AP	4.69	0.00	4.74	0.00	0.98	0.52

Note: SOC, soil organic carbon; TN, total nitrogen; TP, total phosphorus; AN, available nitrogen; AP, available phosphorus.

## Data Availability

Not applicable.

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
