# Peer review of "Impact of Fertilizer on Crop Yield and C:N:P Stoichiometry in Arid and Semi-Arid Soil"

_ijerph, 2021, doi:10.3390/ijerph18084341_

Round 1
Reviewer 1 Report
Neither manuscript nor supplementary materials have figures. Please resubmit with figures. Title can be revised as: “Impact of fertilizer on crop yield and C:N:P stoichiometry in arid and semi-arid soil”. Line 19: Did crop yield patterns were same across the treatments? Line 72: add have: may have effect on or “ may affect, delete on Line 73: may alleviate Line 78 onwards: Keep one line space after subheadings Line 80-81: What is location detection? Is it location for detection and sampling? Line 86 and elsewhere: please revise unit with superscript like m2 Line 90: this study was for only… Line 93-98 and 103-106: Use : after treatment name, CK: then O: then ON: and for others instead of using comma Line 102 and elsewhere: Put Table and figure term as bold Line 102: It would be better if plots can be reversed from right to left instead of left to right Line 103: Not needed here It is already at Line 93-98 Line 114: S type soil sampling (Please cite reference) Line 121: Please revise all chemical formulas with proper subscript Line 139: Table 2: This table should be part of results so please move it after starting result section. Write F-value and P-value in Table subheading. Why some values are not in Bold and others are in Bold? If any specific reason, please explain. Otherwise keep those uniform. Results: Figures are neither in the manuscript nor in the supplementary material.Author Response
Neither manuscript nor supplementary materials have figures. Please resubmit with figures.
Reply: Thank you very much. The figure has been added to the manuscript.
Title can be revised as: “Impact of fertilizer on crop yield and C:N:P stoichiometry in arid and semi-arid soil”.
Reply: Thank you very much. The title has been revised as: “Impact of fertilizer on crop yield and C:N:P stoichiometry in arid and semi-arid soil”
Line 19: Did crop yield patterns were same across the treatments?
Reply: Thank you very much. We have revised the sentence as “Under these treatments except for CK and PK, crop yields initially decreased but later increased. The nutrient content and C:N:P stoichiometry increased in soil depth of 0–20 cm.”
Line 72: add have: may have effect on or “ may affect, delete on
Reply: Thank you very much. We have revised the sentence as “long-term fertilization may effect the crop yield and soil nutrient content.”
Line 73: may alleviate
Reply: Thank you very much. We have revised the sentence as “the changes in soil nutrient content may alleviate the C:N:P stoichiometry of soil.”
Line 78 onwards: Keep one line space after subheadings
Reply: Thank you very much. We have revised it.
Line 80-81: What is location detection? Is it location for detection and sampling?
Reply: Thank you very much. We have revised the sentence as “which is a long-term detection sampling sites.”
Line 86 and elsewhere: please revise unit with superscript like m2
Reply: Thank you very much. We have revised the superscript of all manuscript.
Line 90: this study was for only…
Reply: Thank you very much. We have revised the sentence as “his study was for only selected crops and soil that were planted for millet.”
Line 93-98 and 103-106: Use : after treatment name, CK: then O: then ON: and for others instead of using comma
Reply: Thank you very much. We have revised the sentence into a Table, which can be found in Table 2.
Table 2. Experimental fertilization
|
Treatment |
Illustration |
|
CK |
No fertilizer |
|
O |
0.75 kg/m2 organic fertilizer |
|
ON |
0.75 kg/ m2 organic fertilizer and 0.021kg/ m2 nitrogen |
|
ONP |
0.75 kg/ m2 organic fertilizer, 0.021kg/ m2 nitrogen and 0.017 kg/ m2 phosphorus |
|
OP |
0.75 kg/ m2 organic fertilizer and 0.017 kg/ m2 phosphorus |
|
NP |
0.017 kg/ m2 phosphorus and 0.021kg/ m2 nitrogen |
|
NK |
0.012 kg/ m2 potash and 0.021kg/ m2 nitrogen |
|
NPK |
0.012 kg/ m2 potash, 0.021kg/ m2 nitrogen and 0.017 kg/ m2 phosphorus |
|
PK |
0.012 kg/ m2 potash and 0.017 kg/ m2 phosphorus |
Line 102 and elsewhere: Put Table and figure term as bold
Reply: Thank you very much. We have revised the Table and figures.
Line 102: It would be better if plots can be reversed from right to left instead of left to right
Reply: Thank you very much. The table 1 has been revised.
Table 1 Experimental design of the different fertilization treatments.
|
Experimental plots |
||||||||
|
1 ON |
2 O |
3 NPK |
4 PK |
5 NK |
6 NP |
7 CK |
8 PNK |
9 PP |
|
10 OP |
11 ON |
12 PK |
13 NK |
14 CK |
15 NPK |
16 NK |
17 O |
18 ONP |
|
19 ONP |
20 OP |
21 CK |
22 NK |
23 NP |
24 PK |
25 NPK |
26 ON |
27 O |
|
28 O |
29 ONP |
30 NP |
31 CK |
32 NPK |
33 NK |
34 PK |
35 OP |
36 ON |
Line 103: Not needed here It is already at Line 93-98
Reply: Thank you very much. It has been delected.
Line 114: S type soil sampling (Please cite reference)
Reply: Thank you very much. The reference has been added.
Line 121: Please revise all chemical formulas with proper subscript
Reply: Thank you very much. We have revised the all chemical formulas with proper subscript.
Line 139: Table 2: This table should be part of results so please move it after starting result section.
Reply: Thank you very much. The table3 has been moved after the result.
Write F-value and P-value in Table subheading. Why some values are not in Bold and others are in Bold? If any specific reason, please explain. Otherwise keep those uniform. Results: Figures are neither in the manuscript nor in the supplementary material.
Reply: Thank you very much. The F-value and P-value have been added. Values in the table have been keep uniform. The figure has been added to the manuscript.

Reviewer 2 Report
The topic of the manuscript is interesting, and the authors have presented the research in a compelling narrative. I have some comments and suggestions for changes to improve the manuscript.
L 13-16: Please correct it to ‘To determine the effects of long-term fertilization on crop yield and carbon:nitrogen:phosphorus (C:N:P) stoichiometry in soil a study was conducted on the terraced fields of the Loess Plateau from 2007 to 2019’.
L 26-29: This sentence is too long. Split into 2 sentences or use comma as separator.
L 34: Please provide appropriate reference.
L 72: Please change to ‘..long-term fertilization may affect the crop yield and soil nutrient content….may alleviate….’
L 79: Please attach a map/location of study site to provide geographic clarity to readers.
L 93-99: This section needs to be better represented for clarity to the reader, perhaps as a separate table. The treatments are hard to read and follow in their existing form.
L 114: Please provide appropriate description/reference for sampling method.
Fig 2d, 2e, 3d, 3e: 2010-2013 Please explain why this data is missing, or has not been analyzed.
General comment on placement of figures: The placement of the figures and tables can be improved by moving Fig. 2 after L 160, Fig. 3 after L 164, Fig. 4 after L 170, and Fig 5. after L 175. In the present form, the figures are far away from their citation in the text.
L 311: Fig 6. The correlation matrix is well presented, and the color scheme is helpful. The authors should add a clarification note on what the size of the colored spheres denote in the correlogram.
Author Response
L 13-16: Please correct it to ‘To determine the effects of long-term fertilization on crop yield and carbon:nitrogen:phosphorus (C:N:P) stoichiometry in soil a study was conducted on the terraced fields of the Loess Plateau from 2007 to 2019’.
Reply: Thank you very much. We have revised the sentence as “To determine the effects of long-term fertilization on crop yield and carbon:nitrogen:phosphorus (C:N:P) stoichiometry in soil a study was conducted on the terraced fields of the Loess Plateau from 2007 to 2019.”
L 26-29: This sentence is too long. Split into 2 sentences or use comma as separator.
Reply: Thank you very much. We have revised the sentence as “This study indicated that long-term fertilization can effectively improve crop yield, soil fertility, and soil C:N:P stoichiometry.” And “Meanwhile, the single application of an organic fertilizer or the combination of organic and nitrogen fertilizers can improve the condition of nitrogen limitation in arid and semi-arid areas.”
L 34: Please provide appropriate reference.
Reply: Thank you very much. The reference has been added.
L 72: Please change to ‘..long-term fertilization may affect the crop yield and soil nutrient content….may alleviate….’
Reply: Thank you very much. We have revised the sentence as “long-term fertilization may effect the crop yield and soil nutrient content, (2) the changes in soil nutrient content may alleviate the C:N:P stoichiometry of soil.”
L 79: Please attach a map/location of study site to provide geographic clarity to readers.
Reply: Thank you very much. The figure has been added. We have revised the sentence as “which is a long-term detection sampling sites.”
Figure 1 The locations of study sites in Ansai County, Shanxi Province, China.
L 93-99: This section needs to be better represented for clarity to the reader, perhaps as a separate table. The treatments are hard to read and follow in their existing form.
Reply: Thank you very much. The table has been added.
Table 2. Experimental fertilization
|
Treatment |
Illustration |
|
CK |
No fertilizer |
|
O |
0.75 kg/m2 organic fertilizer |
|
ON |
0.75 kg/ m2 organic fertilizer and 0.021kg/ m2 nitrogen |
|
ONP |
0.75 kg/ m2 organic fertilizer, 0.021kg/ m2 nitrogen and 0.017 kg/ m2 phosphorus |
|
OP |
0.75 kg/ m2 organic fertilizer and 0.017 kg/ m2 phosphorus |
|
NP |
0.017 kg/ m2 phosphorus and 0.021kg/ m2 nitrogen |
|
NK |
0.012 kg/ m2 potash and 0.021kg/ m2 nitrogen |
|
NPK |
0.012 kg/ m2 potash, 0.021kg/ m2 nitrogen and 0.017 kg/ m2 phosphorus |
|
PK |
0.012 kg/ m2 potash and 0.017 kg/ m2 phosphorus |
L 114: Please provide appropriate description/reference for sampling method.
Reply: Thank you very much. The reference has been added.
Fig 2d, 2e, 3d, 3e: 2010-2013 Please explain why this data is missing, or has not been analyzed.
Reply: Thank you very much. The reason of data is missing has been added into the note.
General comment on placement of figures: The placement of the figures and tables can be improved by moving Fig. 2 after L 160, Fig. 3 after L 164, Fig. 4 after L 170, and Fig 5. after L 175. In the present form, the figures are far away from their citation in the text.
Reply: Thank you very much. The location of all figures and table have been changed.
L 311: Fig 6. The correlation matrix is well presented, and the color scheme is helpful. The authors should add a clarification note on what the size of the colored spheres denote in the correlogram.
Reply: Thank you very much. The correlation value has been added into the figure.
Figure 7 Correlation matrix among the different parameters determined for soil depths of 0–20 cm (a) and 20–40 cm (b). Notes: ×, correlation is non-significant at P < 0.05.
Round 2
Reviewer 1 Report
The paper has been revised with suggested revisions and figures included. English construct of line can be improved.